# Exploring the Molecular Basis of Vesicular Stomatitis Virus Pathogenesis in Swine: Insights from Expression Profiling of Primary Macrophages Infected with M51R Mutant Virus

**DOI:** 10.3390/pathogens12070896

**Published:** 2023-06-30

**Authors:** Lauro Velazquez-Salinas, Gisselle N. Medina, Federico Valdez, Selene Zarate, Shannon Collinson, James J. Zhu, Luis L. Rodriguez

**Affiliations:** 1Plum Island Animal Disease Center (PIADC), Agricultural Research Service, USDA, Greenport, NY 11944, USA; 2National Bio and Agro-Defense Facility (NBAF), ARS, USDA, Manhattan, KS 66502, USA; 3Oak Ridge Institute for Science and Education (ORISE)-PIADC, Oak Ridge, TN 37831, USA; 4Posgrado en Ciencias Genómicas, Universidad Autónoma de la Ciudad de Mexico, Ciudad de Mexico 04510, Mexico

**Keywords:** microarray, vesicular stomatitis, porcine macrophages, interferon, M51R mutant, immune response, pathogenesis, RNA sensors

## Abstract

Vesicular stomatitis virus (VSV) is an emergent virus affecting livestock in the US. Previously, using a recombinant VSV carrying the M51R mutation in the matrix protein (rNJ0612NME6-M51R), we evaluated the pathogenesis of this virus in pigs. Our results indicated that rNJ0612NME6-M51R represented an attenuated phenotype in in-vivo and in ex-vivo in pig macrophages, resembling certain clinical features observed in field VSV isolates. In order to gain more insight into the molecular basis leading to the attenuation of rNJ0612NME6-M51R in pigs, we conducted a microarray analysis to assess the gene expression profiles of primary porcine macrophages infected with rNJ0612NME6-M51R compared to its parental virus (rNJ0612NME6). Our results showed an overall higher gene expression in macrophages infected with rNJ0612NME6-M51R. Specifically, we observed that the pathways related with immune cytokine signaling and interferon (IFN)-related responses (including activation, signaling, induction, and antiviral mechanisms) were the ones comprising most of the relevant genes identified during this study. Collectively, the results presented herein highlight the relevance of type I interferon during the pathogenesis of VSV in pigs. The information generated from this study may represent a framework for future studies intended to understand the molecular bases of the pathogenesis of field strains in livestock.

## 1. Introduction

Vesicular stomatitis (VS) is an arboviral disease that affects livestock [1,2]. The disease is caused by the vesicular stomatitis virus (VSV), which is a negative-sense single-stranded RNA virus, and a prototype of the Vesiculovirus genus in the Rhabdoviridae family [3]. This genus encompasses a total of 19 species, among them, Vesiculovirus New Jersey (named as vesicular stomatitis New Jersey virus, or VSNJV) and Vesiculovirus Indiana (vesicular stomatitis Indiana, or VSIV) are responsible for the majority of clinical cases of VS in livestock in the Americas [4]. The genome of VSV encodes five structural proteins: nucleoprotein, phosphoprotein, matrix protein, glycoprotein, and a large polymerase. The matrix protein is one of the most-studied VSV proteins, and is known for its function as a key regulator of the innate immune response [5,6,7].

While advances in our understanding of VSV biology have led to the development of recombinant VSVs with oncolytic properties [8], and as a viral-vectored vaccine platform [9,10], there are still important aspects of the virus that remain poorly understood. Understanding the molecular mechanisms underlying VSV pathogenesis is critical for grasping the extraordinary capacity of the virus to cause cyclic emergence and produce large epidemic outbreaks in the USA [11,12,13,14], making it highly relevant for the livestock industry.

VSV infection in livestock arises through biting insects that carry the virus [15], or by direct contact with an infected animal [16]. The infection causes a disease that is characterized by the development of vesicular lesions in the oral mucosa, tongue, teats, and coronary bands of affected animals [17]. Given their high susceptibility to VSV infection, pigs, which are natural hosts of the virus, have been widely used as an experimental model for studying VSV pathogenesis [16,18,19,20,21,22,23]. The pig model can also be utilized to predict the virulence of VSV (field strains or recombinant viruses) by assessing its ability to induce secondary vesicular lesions in pigs after an initial inoculation via scarification in the snout or direct animal contact [16,23], along with monitoring typical parameters of virulence such as the development of fever and RNAemia [16]. Based on these parameters, it has been typically assumed that VSNJV is more virulent than VSIV due to differences in viral transmission via direct contact and the development of secondary vesicular lesions serving as the main markers of virulence [24]. However, a recent study has shown that certain VSIV strains are comparable in virulence to the VSNJV strains [19]. This suggests that specific genetic determinants among VSIV strains may mediate virulence. Furthermore, a pathogenesis study comparing the virulence between the epidemic strain (NJ0612NME6) and its closest endemic relative, NJ0608VCB, showed that epidemic strains of VSNJV exhibit a more virulent phenotype than endemic ones [16]. Interestingly, the increased virulence observed in NJ0612NME6 was correlated not only with an increased number of vesicular lesions, but also with decreased levels of type I interferon (IFN) in the infected pigs. This observation highlighted the ability for some VSV field strains to suppress the type I IFN response, and represents a potential marker of virulence in pigs.

Recently, we genetically engineered the highly virulent epidemic strain NJ0612NME6 to carry a M51R mutation in the M protein (rNJ0612NME6-M51R) [23]. This mutation has been demonstrated to disrupt the virus’s ability to block the nuclear–cytoplasmic transport of mRNA, which is required for type I IFN production in infected cells [5,25,26]. Interestingly, proof-of-concept studies conducted in swine resulted in an attenuated phenotype. The infection produced vesicular lesions at the inoculation site, but the capacity to produce secondary vesicular lesions and other clinical signs of virulence was decreased (Figure 1). The decreased replication capacity of rNJ0612NME6-M51R in pigs was found to be correlated with reduced viral replication in cultured primary porcine macrophages [23], suggesting an important role for these cells in VSV pathogenesis. Our recent study provides additional evidence for the critical role of porcine macrophages in VSV pathogenesis. Specifically, we identified a subset of differentially expressed genes (DEGs) involved in various immune pathways between primary swine macrophages infected with the field isolate and mock infection in vitro, shedding light on the molecular mechanisms underlying VSV infection [27].

In this study, we aimed to compare host mRNA expression profiles in primary porcine macrophages infected with either the highly virulent strain (WT) or the attenuated strain (M51R) of VSV. Our objective was to identify DEGs and associated biological pathways that may provide insights into the molecular basis mediating the virulence of VSV in pigs. The analysis of our study unveiled several immune-related pathways that were significantly over-represented and predominantly linked with type I interferon (IFN) responses. These findings deepen our understanding of VSV pathogenesis, and could facilitate future in vitro studies to characterize VSV field strains.

## 2. Materials and Methods

### 2.1. Cells and Viruses

Primary swine macrophages were prepared from defibrinated blood, as previously described [28]. The field isolate virus, rNJ0612NME6, and recombinant virus, rNJ0612NME6-M51R, were obtained from the high titer viral stocks produced in Vero cells during a previous study [23]. Both viruses were derived from our infectious clone system [29]. In the case of rNJ0612NME6-M51R, mutation at M51R in the matrix protein was engineered via site-directed mutagenesis, as previously described [23].

### 2.2. RNA Extraction

Infections with rNJ0612NME6 and rNJ0612NME6-M51R were carried out on primary swine porcine macrophages using three independent biological replicates collected from three pigs. Six-well plates containing 1 × 10^7^ cells per well were infected at an MOI of 10 TCID50 with each virus, and incubated for five hours at 37 °C under 5% of CO_2_. The infections were conducted in triplicate. The mock-infected samples were incubated in the same conditions with macrophage media in triplicate. The incubation time was determined based on previous studies performed on primary porcine macrophages [22,23,27]. It is supported by the ability of VSV to induce high levels of gene expression in primary porcine macrophages infected with a high MOI, avoiding the presence of the cytopathic effect, a condition that decreases the RNA yields. After incubation, the total RNA was extracted using the RNeasy mini kit (Qiagen LLC-USA, Germantown, MD, USA cat# 74004) according to the manufacturer’s instructions. RNA quality was assessed using the Agilent 2100 bioanalyzer (Santa Clara, CA, USA), using an RNA nanochip. The quantification of RNA was conducted on a Nanodrop 1000 (Thermo Scientific, Rockwood, TN, USA).

### 2.3. ELISA

A commercial enzyme-linked immunosorbent assay (ELISA) (Abcam porcine IFN-alpha ELISA kit, Cat #ab273213) was used to compare the abilities of rNJ0612NME6 and rNJ0612NME6-M51R to induce the secretion of porcine type I IFN (IFNα) in primary swine porcine macrophages. For this purpose, six-well plates containing 1 × 10^7^ cells per well were infected at an MOI of 10 TCID50, with each virus and incubated for five hours at 37 °C under 5% of CO_2_. Supernatants were collected and evaluated via ELISA, following the manufacturer’s protocol. The experiments were conducted in three biological replicates.

### 2.4. DNA Microarray Analysis

Microarray analyses were conducted using a previously described porcine microarray [27,30]. The porcine microarrays were custom designed and provided by Agilent Technologies. The RNA samples were labeled with Cy3 and Cy5 individually, using a low-input RNA labeling kit (Agilent Technologies cat# Part # 5190-2306). The Cy5-labeled RNA samples were co-hybridized with the corresponding Cy3-labeled sample (e.g., VSV and mock-infected) using a dye-swap design. Array slides were scanned using a GenePix 4000B scanner (Molecular Devices) with the GenePix Pro 6.0 software (Molecular Devices) at 5 μM resolution. The procedure and analysis were conducted according to manufacturers’ instructions.

The LIMMA package included in the R software was used to conduct data normalization, background correction, and statistical analysis [31]. Log2-fold changes in signal intensity were used to identify differentially expressed genes between the VSV and mock-infected samples. The Benjamini and Hochberg method was used to adjust *p*-values that were ultimately expressed as a false discovery rate (FDR). In this sense, genes showing FDR values of ≤0.05 and an expression difference of ≥50% were considered to be significant differentially expressed genes (DEGs) between the VSV-infected and mock-infected samples. A validation with qPCR of some of the DEGs reported in this study is provided in Appendix A. This experiment was conducted using three biological replicates, following a methodology previously described [23].

### 2.5. Hierarchical Cluster Analysis

Hierarchical cluster analysis (HCA) was used to categorize the correlation of selected DEGs between rNJ0612NME6 and rNJ0612NME6-M51R. HCA uses a matrix of N × M dimensions, where N is the number of DEGs and M is the number of viruses used in the analysis. The overall data were robustly standardized, and clusters were defined by the complete linkage method. Once the number of clusters was defined, an analysis of variance (ANOVA) supported by Tukey’s honest significance test (a = 0.05) was conducted to identify significant clusters in terms of the differential in the levels of gene expressions between rNJ0612NME6 and rNJ0612NME6-M51R. The analyses were conducted using the predictive analytic software JMP^®^ Pro version 16.0.0.

### 2.6. Pathway Analysis

Pig gene sequences of microarray probes were mapped to human genes using BLAST and/or manually annotated with genetic information displayed on the UCSC Genome Browser website. Human gene Entrez-ID numbers associated with DEG were used in the pathway analysis using the database for annotation, visualization, and integrated discovery (DAVID), v6.8 updated 2021 version [32]. The biological pathways were identified using the reactome pathway analysis [33], considering an FDR of ≤0.05 as the significant threshold, and the human database as a model for the analyses.

### 2.7. Euler Diagram

An Euler diagram was used to represent the relationship among genes of multiple identified pathways. This diagram is frequently used for the analysis of microarray experiments [34], and highlights relevant genes shared by different pathways reflecting potential functional associations between them. These associations are represented by closed circles that split the plane into divided but connected subcategories [35]. The analysis was conducted using the R package eulerr.

### 2.8. Biological Inference

The biological functions of the identified DEGs in the identified in over-represented pathways associated with the immune response were inferred based on scientific publications literature obtained from PubMed (https://pubmed.ncbi.nlm.nih.gov/ accessed on 1 January 2023). The biological inferences were based on (i) the immunological functions of the DEGs, (ii) the signal intensities of gene expression levels based on microarray-averaged signal intensities, and (iii) magnitudes (fold) of the differential expressions, assuming that higher mean signal intensities and therefore larger differentially expressed genes play a more significant bigger biological role in the gene groups. Genes with no significant differential expression, but that are known to play important roles in the biological pathways associated with the significant DEGs, but without differential expression themselves, were used as supporting evidence. Genes that were down- or up-regulated in the VSV-infected samples compared to the mock-infected samples were expressed as negative and positive values (fold), respectively. In this study, the genes that were differentially expressed between infected and mock-infected macrophages were used to further infer the molecular mechanisms of VSV pathogenesis and immune evasion.

## 3. Results

### 3.1. Clustering Analysis of DEGs

A total of 4024 DEGs (up- or down-regulated) between macrophages infected with either rNJ0612NME6 or rNJ0612NME6-M51R and the mock-infected cells were detected. HCA was used to group these DEGs into 20 clusters (Figure 2A), with cluster numbers 2 and 15 comprising 70.6% of the total DEGs. In these two clusters, the mean differential expressions between the two infected samples were only 0.77- and 0.43-fold, respectively, indicating that most of the identified genes were similarly expressed (Figure 2B). In contrast, a statistically significant difference between the two infected samples was found in six clusters (4, 9, 10, 11, 13 and 17) identified by ANOVA, which was further confirmed by the Tukey’s honest significance test (a = 0.05). These clusters displayed the highest differential expression means between the two infected samples compared to the other clusters. These six clusters consisted of a total of 106 genes (2.63% of the DEGs), with the differential expression means ranging from 6.13- and 30.19-fold (Figure 2B). In addition, another group of relevant clusters (1, 3, 5, 12, 14, 16, 20) comprising a total of 718 DEGs was identified, with differential mean gene expressions ranging from 2.20- to 3.98-fold between the two infected samples. Given their significant differential expression profiles, these two groups of clusters were further analyzed using pathway analysis. The remaining clusters containing a mean differential expression value lower than 2.20-fold were excluded from the identification pathway analysis (clusters 2, 6, 7, 8, 15, 18, 19).

### 3.2. DEG Pathway Analysis

The DAVID functional annotation tool was used to analyze the 13 selected gene clusters (1, 3–5, 9–14, 16, 17, 20), in order to identify the biological pathways that were significantly affected by differential expression. Our analysis revealed 19 pathways that were identified using the reactome pathway classification analysis with an FDR of <0.05 (Figure 3). All of the pathways were immune-related, with 166 out of the 824 DEGs containing multiple DEGs redundantly involved in different immunological pathways. Among these pathways, 15 contained 105 DEGs, and were statistically significant in terms of the differential mean expression between rNJ0612NME6- and the rNJ0612NME6-M51R-infected samples, as determined by unpaired *t*-tests (*p* < 0.05) (Figure 3, highlighted in grey). Fourteen pathways were associated with specific immune responses. Among them, the immune cytokine signaling pathway was found to be the most significantly affected, followed by eight pathways involved in IFN-related responses including activation, signaling, induction, and antiviral mechanisms (Figure 4). IFN-related pathways comprised a total of 56 genes accounting for 53% of the DEGs. The other five pathways were associated with MHC antigen processing and presentation or regulation of cell death (Figure 4). Interestingly, the Euler diagram showed 18 genes that were differentially upregulated among the different biological pathways identified in this study. Of relevance, the intracellular nucleic acid sensors DDX58 and IFIH1 (also known as RIG-I and MDA5, respectively) may play a central role in mediating the induction of type I IFN (alpha/beta) pathways, and consequently the up-regulation of other biological pathways (Figure 5). Our analysis suggests that the activation of viral RNA sensors during VSV infection may be a critical step in the control of this virus in a porcine model. However, given the functional redundancy among sensors, adaptors, and effectors of the IFN signaling pathway, defining a single functional network of genes responding to VSV infection may be challenging. A more detailed explanation about the multiple DEGs involved in different pathways identified in these analyses is presented in the following sections.

### 3.3. DEGs Associated with IFN Responses

Macrophages infected with the M51R mutant virus expressed significantly higher levels of type I (IFNA, IFNB, IFND, and IFNW) and type III (IFNL) IFNs, but not type II IFN (IFNG) (Figure 6A). Among these IFNs, IFNA genes (including 2, 4, 6, and 17) were the most differentially expressed, followed by IFNL and IFND, where increased expression was observed in macrophages infected with the mutant virus compared to the WT virus-infected cells. The mutant-infected macrophages also expressed significantly higher levels of three IFN receptors (IFNAR1, IFNAR2, and IFNGR1), one IFN signaling transducer (JAK2), and three transcription factors (IRF9, STAT1, and STAT2) compared to the WT-infected cells (Figure 6B). These results suggest that the mutant-infected macrophages not only transcribed more type I and III IFNs, but also transduced IFN signals better than the WT-infected macrophages. This idea was strongly supported by the significantly up-regulated expression of many IFN-stimulated genes (ISGs) in the mutant-infected macrophages (Figure 4A). The results indicate that the M51R mutation in the M protein reduced the VSV’s ability to suppress IFN production and signaling. To corroborate with the results obtained by microarray analysis, we conducted type I IFN ELISA and showed that the type I IFN concentration was 3.77 times higher in the supernatants from cell culture infected with the mutant virus (1478.53 ± 86.89 pg/mL vs. 391.64 ± 73.62 pg/mL) than in those from WT virus infection.

Additionally, the expressions of three cytosolic pattern recognition receptors, IFIH1, RIG-I, and ZNFX1 [36,37], and an enhancer (TRIM25) [38], were significantly up-regulated in mutant-infected macrophages when compared with those infected with the WT virus (Figure 6C). In contrast, WT infection induced significantly higher expressions of genes that negatively regulate type I IFN expression, such as AHR [39], DUSP1 [40], HES1 [41], and PRDM1 [42], compared to the mutant-infected cells (Figure 6D). The expressions of these three genes were comparable between the mock-infected and mutant-infected macrophages (Appendix A), indicating that the observed differences were caused by the WT infection. IRF3 and IRF7 are transcription factors that induce IFNB and IFNA transcription, respectively [43,44] (Figure 6E). FOS and FOSL2 are anti-inflammatory transcription factors that suppress NFκB activity [45], and may also suppress IFN expression. These genes were significantly down-regulated in the mutant virus-infected macrophages compared to the WT-infected cells (Figure 6F). Differential expression was observed for IRF7 but not for IRF3 (Figure 6E), which may explain the observed increased differential expression of IFNA genes compared to that of IFNB between the mutant- and WT-infected macrophages.

We also identified several DEGs that exhibit suppressive effects on IFN or RIG-I-like signaling in macrophages infected with the M51R mutant virus. These DEGs included three genes (SOCS1, SOCS3, and USP18) [46] (Figure 6G) involved in suppressing IFN signaling, and six genes (IRF2, NLRC5, TRIM13, TRIM 21, TRIM26, and TRIM38) [46] involved in suppressing RIG-I-like signaling, which were significantly up-regulated (Figure 6H).

### 3.4. DEGs of Interleukins, Chemokines, and Their Receptors

Several cytokines were up-regulated during the infection with both viruses (Figure 7A). Interestingly, ILIRN, known for its role to modulate inflammatory functions of cytokines including those stimulated by IL1A and IL1B [47], correlated with the clinical outcome linked to reduce disease severity in animals inoculated with this mutant virus. Furthermore, this suggests that the mutant virus may not induce an increase in body temperature, as was observed in the WT infection phenotype. Moreover, we found a reduction in CSF2 expression in the mutant-infected cells compared to WT-infected cells (Figure 7A). CSF2 is known to act as an anti-inflammatory/regulatory cytokine, promoting tolerogenic dendritic cell differentiation to enhance the numbers and function of regulatory T-cells [48]. The expression of IL15 (NK and cytotoxic T-cell-activating cytokine) [49] was not significantly different between both viruses. However, IL18 (Th1 response-enhancing cytokine) [50], IL27 (a T regulatory cell-promoting cytokine) [51], and TNFSF13B (a B-cell-activating TNF family member) [52] were significantly up-regulated in the mutant-infected macrophages compared to levels in WT-infected cells (Figure 7A). LIF, an anti-inflammatory cytokine [53] and a potent inhibitor of feed intake [54], was expressed at a significantly lower level in the mutant virus-infected cells compared to the WT-infected cells (Figure 7A). The expressions of IL1B (an endogenous pyrogen and proinflammatory cytokine) and IL10 (an immune suppressive cytokine) were not significantly different between the mutant- and WT-infected macrophages (Figure 7A).

Our analysis identified eight interleukin receptors that were differentially expressed between the mutant- and WT-infected cells (Figure 7B). Notably, the expression of IL1R2 was down-regulated in mutant-infected cells compared to WT-infected cells, while IL1RAP expression was slightly up-regulated by the mutant infection. IL1R2 is a decoy receptor of IL1A and IL1B that interacts with IL1RAP to inhibit IL-1 signaling [55]. These results suggest that infection with the mutant virus may increase IL-1 signaling. The high-affinity receptor of IL15, IL15RA, can trans-present IL15 to potently stimulate NK and CD8+ Teff cells [56]. This IL15 receptor was significantly up-regulated by the mutant-infected cells. Among the other four up-regulated cytokine receptors, IL7R (stimulation of B-cell maturation, T and NK cell survival) and IL10RA (the receptor of the suppressive cytokine, IL-10) were up-regulated in the mutant-infected cells compared to the WT-infected cells (Figure 7B).

Different CCLs were expressed during the infection with both viruses (Figure 7C). Overall, higher levels of expression were seen in CCL20 and CCL4L, being slightly higher in the cells infected with the mutant virus (Figure 7C). Moreover, slightly higher levels of expression were observed in CCL5 during the infection with the mutant virus. Only the expression of CCL8 was significantly increased in comparison to the WT-infected cells (Figure 7C). CCL4 and CCL5 recruit macrophages and T cells [57]. CCL20 is a chemotactic factor for Th17 cells, and CCL8 promotes the Th2 response [57]. Regarding the overall expression of ELR+ (glutamic acid-leucine-arginine) CXCLs that recruit neutrophils and CD8+ effector T cells, the WT-infected cells expressed higher levels of all ELR+CXCLs than the mutant virus-infected cells (CXCL5, CXCL3, and CXCL2) (Figure 7D). The total expression of three ELR-CXCLs (CXCL9, CXCL10 and CXCL11) that recruit Th1, CD8+ T cells, and NK cells were up-regulated in mutant-infected cells compared to mock-infected cells (Figure 7E). Furthermore, CXCL13 was expressed slightly more in mutant-infected cells compared to WT-infected cells (Figure 7E).

The expressions of CCR5 (the receptor of CCL3, CCL4, and CCL5) and CCRL2 (expressed on neutrophils and primary monocytes) were significantly down-regulated by the WT infection (Figure 7F). Therefore, the expressions of CCR5 and CCLR2 were shown to be increased by the mutant-infected cells (Figure 7F). On the other hand, the expressions of CCR7 and CXCR4 were significantly induced by WT infection.

### 3.5. DEGs in MHC Antigen Presentation

Efficient MHC class I antigen presentation involves distinct biological pathways including ubiquitination, proteasomal degradation, and peptide loading assembly. In this study, the expression profiles of transcripts linked to ubiquitination (i.e., AREL1, RBBP6, UBA7, UBE2A, UBE2D1, UBE2H, UBE2B, UBE2K, UBE3A) and proteasomal degradation were up-regulated in cells infected with the VSV mutant when compared to mock infection (Figure 8A,B). Among the three peptide transporters identified, an increased expression was observed in TAP1 and TAP2 genes in cells infected with the mutant virus, while a higher expression was recorded for TAPBP during infection with the WT virus (Figure 8C).

In MHC class II antigen presentation, antigen acquisition and processing heavily rely on the endosomal/lysosomal compartments containing specific proteases such as cathepsins [58]. Two MHC class II-processing enzymes (CTSC and CTSH) were expressed at slightly higher levels, although only the increased expression of CTSC was statistically significant in the mutant-infected cells (Figure 8D). Conversely, higher expression levels were seen in the CTSB gene in cells infected with the WT virus (Figure 8D). Interestingly, TNFSF18 (a Th17-, Tfh-, and Th9 cell-activating TNF family member) [59] and CD40 (a T cell activation co-receptor of antigen presentation cells) [60] were significantly up-regulated by the mutant-infected macrophages when compared to WT-infected cells (Figure 8E). These results suggested that the mutant-infected cells could present MHC class I and II peptides more efficiently than the WT-infected cells.

### 3.6. DEGs in Apoptosis

VSV is well known for its capacity to induce apoptosis in infected cells [61]. Our results were consistent with previous publications indicating the increased and rapid ability of the VSV M51R mutation at the matrix protein to induce apoptosis in comparison with the WT protein [62]. Thirteen pro-apoptotic genes were expressed at higher levels by mutant-infected cells than the WT-infected cells (Figure 9A). The expressions of two death receptor (DR) ligands, TNFSF10 and TNFSF15, were higher in the mutant-infected cells (Figure 9B). In contrast, pro-apoptotic factor TNF was up-regulated in both infections, but not differentially expressed between the infections (Figure 9B). Three death receptors (TNFRSF1A, FAS, and TNFRSF10) were expressed at higher levels after mutant infection. Another TNF receptor, but not a death receptor, TNFRSF3/LTBR, was also up-regulated, whereas TNFRSF1B (an anti-apoptotic receptor) [63] was down-regulated. Three DR-signal-transducing genes (FADD, RIPK1, and TRAF2) were expressed at significantly higher levels in mutant-infected cells (Figure 9B), whereas the expressions of anti-apoptotic genes (REL and TNFRSF1B) in DR signaling pathways were down-regulated by mutant-infected cells compared to WT-infected cells (Figure 9B,C). These results suggest that macrophages infected with the mutant virus were able to express death receptor ligands at higher levels, and may be more sensitive to DR ligand-induced cell death or apoptosis.

### 3.7. Potential Relevance of the DGEs Observed in This Model

The effects of the M51R mutation in the VSV matrix protein inferred from differential gene expression are summarized in Table 1 The attenuation of M51R mutation in pigs is likely mediated by the up-regulated expressions of mainly type I and III IFN, IL1RN, and TNFSF10, and the down-regulated expression of LIF in the non-infected tissues. The up-regulated expressions of ELR+ CXCLs, IL1R2, death receptor, and their ligands could play additional roles in the control of inflammation and VSV infection of infected tissues. The genes induced by the mutant virus may also enhance MHC antigen processing and presentation as well as the immune response to induce early onset of protection via the innate antiviral activity of IFN, suggesting that the mutant virus could serve as a vector vaccine candidate. Additionally, the M51R mutation may also improve VSV oncolytic activity (Table 1).

## 4. Discussion

The identification of molecular mechanisms associated with the virulence of VSV is critical to understand its pathogenesis in livestock. Currently, most of the knowledge in terms of its pathogeny comes from VSV mice models where macrophages, especially those residing in the lymph nodes, act as key players in the immune response against VSV [64,65,66,67,68]. In this regard, we used primary cultures of swine macrophages to evaluate immunological responses during VSV infection. In terms of host gene expression, VSV M protein inhibits the transcription of target genes [69] and nucleocytoplasmic transport [70]. Human IFNB gene is one of the genes that is targeted by the M protein [25]. The attenuation by mutations in the M protein have been well-characterized in mice [71]. We created the recombinant VSV with a similar mutation (rNJ0612NME6-M51R) as an experimental model for study in pigs. The rNJ0612NME6-M51R mutation was grown in ex vivo swine macrophages and displayed attenuated phenotypes in pigs [23].

Although rNJ0612NME6-M51R can induce large amounts of type I IFN in other primary cultures such as fetal kidney porcine cells (a mix of fibroblasts and epithelial cells), the growth of this virus was not compromised when compared with the parental strain. This observation is consistent with the ability of rNJ0612NME6-M51R to cause epithelial lesions upon intradermal inoculation in the snouts of pigs [23], highlighting the significance of macrophages in the pathogenesis of VSV in swine. In this study, we contrasted the genetic expression profiles between primary swine macrophages infected with the epidemic and highly virulent strains of VSNJV rNJ0612NME6 and the attenuated M51R mutant, in order to understand the mechanisms of attenuation.

We recently reported on genes induced by primary swine macrophages infected with the field isolate to infer VSV molecular pathogenesis [27]. In this study, our biological pathway analyses of DEGs identified several pathways associated with the type I IFN response (including activation, signaling, induction, and antiviral mechanisms), which was consistent with previous in vitro and in vivo experiments that demonstrated the susceptibility of VSV to the action of type I IFN [22,72,73,74]. Our findings indicate that the expressions of type I IFNs were significantly elevated in the mutant-infected cells compared to those in the WT-infected cells, confirming our expectations. Interestingly, we also observed a significant increase in the expressions of type III IFNs (IFNL), which has not been previously reported. These observations highlight the importance of further investigating the role of type III IFNs in the antiviral response to VSV infection.

In this study, the expression of IFNB in both mutant- and WT-infected cells was up-regulated. The studies of IFNB gene promoter have shown that activated ATF2-JUN, IRF3, and NFκB transcription factors induced IFNB expression [75]. These transcription factors were not differentially expressed between the M51R mutant and WT infection in this study. Instead, four IFN expression inhibitory genes (AHR, DUSP1, HES1, and PRDM1) were significantly up-regulated by WT infection. AHR down-regulates the IFNB production via up-regulating the expression of TIPARP to inhibit the activity of TBK1, a key signal transducer of IRF3 and NFκB signaling [39]. DUSP1 and HES1 are the inhibitors of pattern recognition receptor signaling pathways. DUSP1 is a negative regulator of mitogen-activated protein kinases (MAPK) that can inhibit IFNB expression by macrophages [40]. HES1 suppresses IFNB expression via attenuating signaling of poly(I:C), a synthetic mimic of viral double-stranded RNA, by inhibition of an adaptor molecule, WDFY1 [41]. PRDM1 is a transcription repressor that binds to the IFNB gene promoter [41]. Additionally, two other genes (FOS and FOSL2) were significantly up-regulated in WT-infected cells, but not in mutant-infected cells. These genes are anti-inflammatory-related transcription factors that suppress NFκB activity [45]. FOS and JUN form heterodimers called AP-1 [39], and a high expression of FOS could compete with ATF2 for dimerization, thereby suppressing IFNB transcription. The up-regulated expressions of these six genes (AHR, DUSP1, HES1, PRDM1, FOS, and FOSL2) in the WT- but not the mutant-infected cells suggest their role in suppressing IFNB expression by the M protein. In fact, it has been previously reported that the VSV M protein inhibits NFκB activation [76]. Our results suggest more detailed mechanisms of the M protein in suppressing IFNB expression.

IRF7 is the critical transcription factor of type I IFN expression induction, and the constitutive and induced expression of IRF7 in macrophages is primarily controlled by IFNB [44]. IRF7 expression was significantly up-regulated by the M51R mutant infection compared to the mock infection, whereas its expression between WT and the mock infection was comparable. Therefore, the suppressed production of type I IFNs by the VSV M protein was most likely mediated by suppression of IFNB expression via up-regulation of these six IFN inhibitory genes, including FOS and FOSL2. Additionally, our results revealed that the WT virus was capable of suppressing type I IFN signaling through down-regulation of IFNR2, IRF9, and STAT2 expression, which could subsequently limit the downstream effects of IFNB in inducing the expression of other type I IFNs. These differentially expressed genes provide insight into the mechanisms underlying the suppression of type I and possible type III IFN responses by WT-infected cells.

Given the potent antiviral activity of type I and III IFNs and our results, it is likely that the attenuation of VSV M51R in pigs is due to the reduction in the suppressive effects of the VSV M protein on IFNB expression and signaling. The expressions of IL1RN and TNFSF10 in mutant-infected cells were 22.7- and 80.5-fold higher, respectively, than those in WT-infected cells. IL1RN is the antagonist of IL-1 cytokines (potent pyrogens) [77]. The high induction of IL1RN in mutant-infected but not in WT-infected cells could be the main contribution to the lack of fever observed in the mutant-infected pigs [23]. Since IFN signaling induces IL1RN expression [78,79], the up-regulated IL1RN was likely due to increased expression of type I IFN. Likewise, human TNFSF10 induces cell death of virus-infected cells but not of uninfected cells, and its expression could also be induced by type I IFN signaling [80]. Therefore, up-regulated TNFSF10 in the mutant infection could only be another factor contributing to the attenuation.

Post-translational modifications (i.e., phosphorylation, ubiquitination, ISGylation) activate transcription factors that modulate IFN transcription. It is worth noting that substantial up-regulation of transcripts (UBA7, UBE2L6, HERC5) encoding cellular functions involved in conjugating the ubiquitin-like molecule ISG15 (ISGylation) to different protein substrates were featured in the Euler diagram. These results suggest that infection with the VSV mutant virus may promote cellular ISGylation to modulate immune responses in macrophages. In support of these observations, recent studies have demonstrated that protein ISGylation may regulate the inflammatory responses of macrophages during virus infection [81]. Further studies measuring total ISGylation are warranted to examine this role in the pathogenesis of VSV.

The activation of apoptosis by rNJ0612NME6-M51R using the extrinsic pathway was highly supported by our findings, identifying the relevance of the death receptor signaling pathway during our analysis according to [82]. TNF represents a key molecule to initiate the activation of the extrinsic pathway [83]. Interestingly, TNF was up-regulated by both the mutant and WT viruses at nearly the same level, by approximately 25-fold. TNF has been described for its essential role in promoting the survival of CD169+ macrophages during the infection of VSV in mice [84], suggesting that the ability of both viruses to induce TNF expression in macrophages was not responsible for the phenotypic differences seen in both viruses in pigs.

The advances in the understanding of the biology of VSV have led to the development of VSV recombinant viruses with oncolytic properties [8], and as a virally vectored vaccine platform [9,10]. In this sense, our results suggest that the M51R VSV mutant may be a better viral vaccine vector than VSV without the M51R mutation, based on the genes induced after infection; a summary of the differential gene expressions between mutant and WT infection is listed in Table 1. Firstly, the mutant infection induced production of a large amount of type I and III interferons, which can translate into early innate immune protection. Secondly, the ubiquitin–proteasome system is critical for MHC antigen processing [85]. Our results showed that the expressions of several ubiquitins were induced by the mutant infection, and the overall expressions of ubiquitin and proteasome protease genes were expressed at higher levels in the mutant-infected macrophages than in the WT-infected cells. Thirdly, several up-regulated genes, including IL18 [50], CCL4, CCL5, CCL8, CCL20 [57], CD40 [60], TNFSF13B [52], and TNFSF18 [59], may enhance MHC antigen presentation or the immune response except IL27, which mainly promotes the Treg response [51]. Likewise, the M51R VSV mutant may be a better oncolytic virus than the VSV without the M51R mutation. Type I IFN [86] and TNFSF10 [80] have been reported to have antitumor activity, and their expressions were significantly higher in the mutant-infected cells than the WT-infected cells.

## 5. Conclusions

Collectively, our results highlight the potential relevance of type I IFN-involved pathways in the pathogenesis of VSV in pigs. This assertion is supported by our previous experiment utilizing VSV-IFN beta-NIS, which demonstrated the impact of type IFN on the pathogenesis in pigs [22]. Our analysis suggested the importance of the DDX58/IFIHI-mediated induction of the interferon alpha/beta and TRAF6-mediated IRF7 activation pathways to explain the differences in pathogenesis between rNJ0612NME6 and rNJ0612NME6-M51R. In this sense, while previous studies have evaluated the induction of type I IFN by various isolates in primary cultures of chicken fibroblasts [87], it is important to emphasize the significance of conducting these experiments using a more physiologically relevant cell line to study the pathogenesis of VSV. Building upon the results presented in this study, we are currently investigating the capacity of an extensive and genetically diverse collection of VSV isolates, representative of multiple geographic regions in the Americas, in order to induce type I IFN in porcine macrophages.

The main goal of this study was to utilize rNJ0612NME6-M51R as proof-of-concept to unravel the molecular basis of VSV pathogenesis in pigs, with the ultimate aim of extending this understanding to other susceptible livestock species. Currently, there is limited characterization of VSV strains in terms of their virulence levels in pigs and other livestock susceptible species. Nevertheless, our findings demonstrate important parallels between the pathogenicity induced by rNJ0612NME6-M51R and some field isolates of VSIV in pigs [18,19,23,24]. Therefore, our study offers a platform for future research aimed at understanding the variations in virulence between VSNJV and VSIV. Finally, our study identified multiple genes implicated in the repression of type I IFN, shedding light on potential avenues for investigating alternative mechanisms of cytokine control. It is important to note that this study primarily serves as a descriptive analysis, and as such, additional experiments are warranted to validate and explore the hypothesis generated herein. Such investigations will contribute to a more comprehensive understanding of the complex interplay between gene regulation and type I IFN responses during VSV infection.

## Figures and Tables

**Figure 1 pathogens-12-00896-f001:**
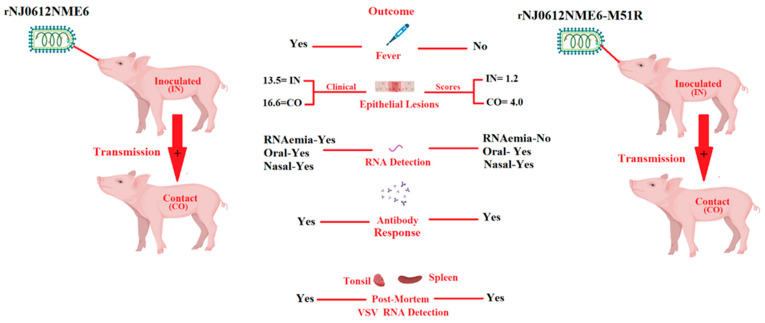
Phenotypic characteristics displayed by rNJ0612NME6-M51R during its infection in pigs. Contrasting clinical scores were observed between pigs infected by direct inoculation (IN) and contact exposure (CO) with rNJ0612NME6 and rNJ0612NME6-M51R, indicating the reduced ability of rNJ0612NME6-M51R to produce the development of secondary vesicular lesions after infection, a situation that resembles some clinical outcomes produced by VSIV strains. The attenuated phenotype showed by rNJ0612NME6-M51R was also correlated with the absence of fever and RNAemia in IN and CO pigs infected with this virus. The presence of viral RNA in the spleen of pigs infected with rNJ0612NME6-M51R indicates the ability of this virus to produce systemic infection. However, in both groups no infectious virus was recovered from post-mortem tissues. Interestingly. The lack of correlation between the presence of RNA in spleen samples from pigs infected with M51R, may be explained based on the decreased number of vesicular lesions produced in these pigs. The correlation between the intensity of the RNAemia and the number of vesicular lesion in infected animals has been previously reported [16]. The relevance of RNAemia in the pathogenesis of VSV, is currently under investigation. Based on these results, we are proposing the use of rNJ0612NME6-M51R as a potential model to obtain more insight into the molecular basis of the pathogenies of VSV in pigs. Data from this figure were obtained from Velazquez-Salinas et al. [23]. This figure was created using BioRender.com, under the academic license number: QX25JBPWHS.

**Figure 2 pathogens-12-00896-f002:**
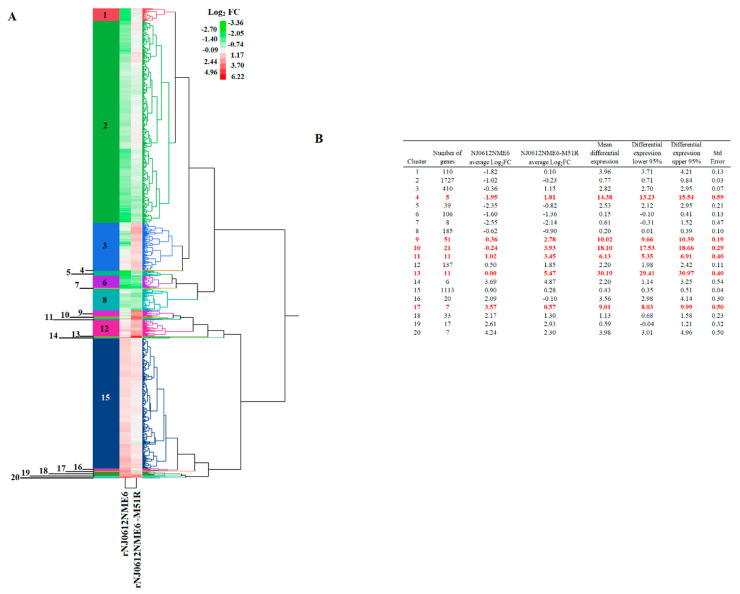
Hierarchical cluster analysis (HCA) of differentially expressed transcripts. (**A**) HCA was conducted to identify relevant groups of genes differentially expressed in cell cultures of primary swine macrophages after infection with either rNJ0612NME6 or rNJ0612NME6-M51R. Levels of expression are displayed in terms of fold change (FC) Log_2_. Negative and positive values reflect genes that were down- or up-regulated, respectively. (**B**) Summary of the statistics displayed by different clusters. Numbers in red represent mean differential expressions that were significantly different from the rest of the clusters. Calculations were determined with ANOVA and supported by Tukey’s honest significance test.

**Figure 3 pathogens-12-00896-f003:**
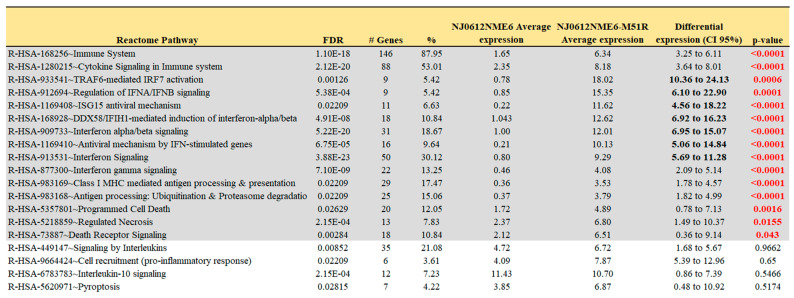
Reactome pathway analysis. Reactome pathway analysis was conducted on a total 824 genes using the DAVID functional annotation tool. Selection of pathways was carried out based on FDR (*p* ≤ 0.05) criteria. Pathways highlighted in grey represent those with a statistically significant differential gene expression (*p* < 0.05) between rNJ0612NME6 and rNJ0612NME6-M51R. Details about the levels of expression of specific genes within the relevant pathways are presented in Figure 4.

**Figure 4 pathogens-12-00896-f004:**
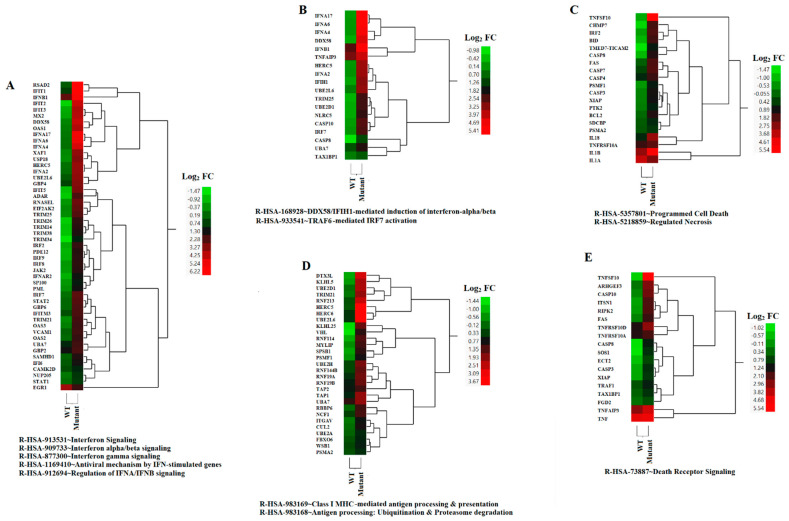
HCA of relevant biological pathways differentially expressed due to infection by rNJ0612NME6 (WT) or rNJ0612NME6-M51R (mutant). Differential levels of expressions within genes associated with the identified biological/cellular pathways are represented (**A**–**E**). Values are expressed in terms of fold change (FC) Log_2_. The spectrum of colors ranges from green to red and reflect down- and up-regulation, respectively.

**Figure 5 pathogens-12-00896-f005:**
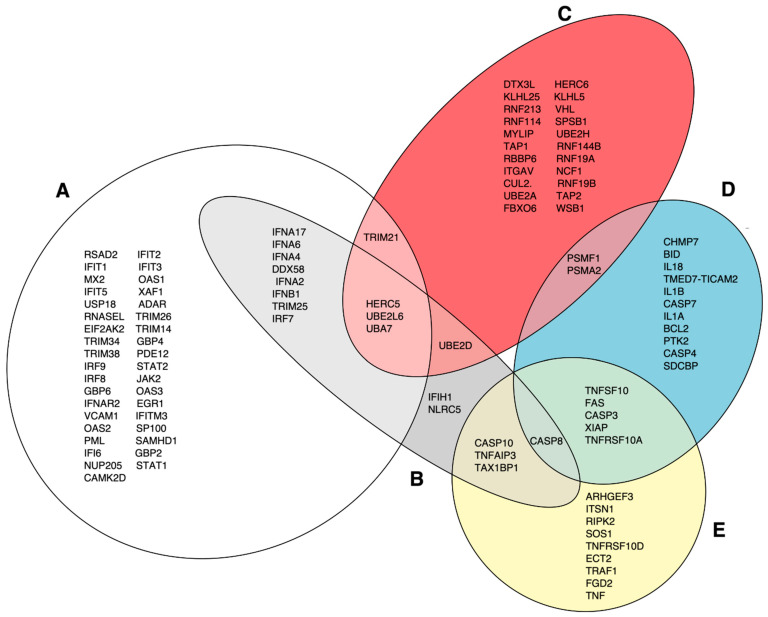
Euler diagram analysis assessing the relationship among different pathways identified in this study. (**A**) represents genes in the following pathways: IFN signaling, IFN alpha/beta signaling, IFN gamma signaling, antiviral mechanisms by IFN-stimulated genes (ISGs), and regulation of IFNA/IFNB signaling. (**B**) DDX58/IFIH1-mediated induction of IFN alpha/beta. (**C**) Class I MHC-mediated antigen and presentation and antigen processing: ubiquitination and proteosome degradation. (**D**) Programed cell death and regulated necrosis. (**E**) Death receptor signaling.

**Figure 6 pathogens-12-00896-f006:**
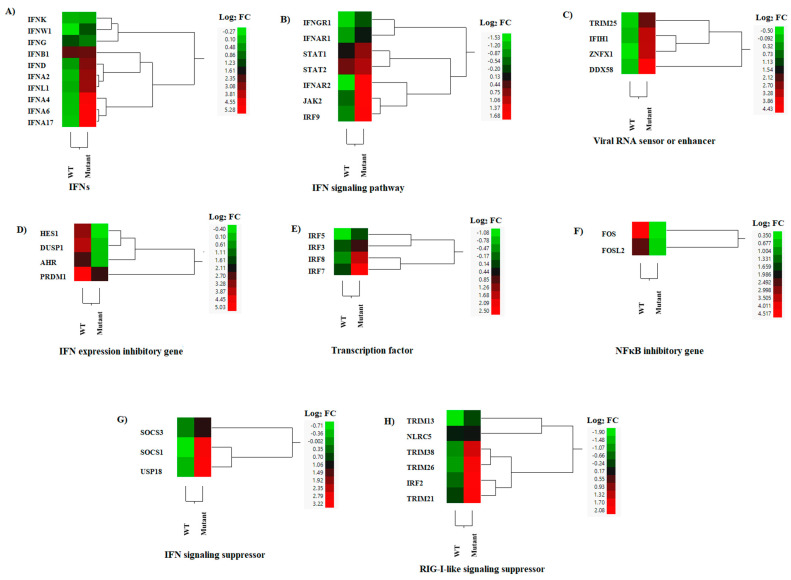
Hierarchical cluster analysis (HCA) showing the differential gene expressions (DGEs) between rNJ0612NME6 (WT) or rNJ0612NME6-M51R (mutant) of genes associated with interferon (IFN) responses. These analysis included genes associated with; (**A**) Interferons (IFNs), (**B**) IFN signaling pathway, (**C**) Viral RNA sensing and enhancement of the IFN response, (**D**) Inhibition of IFN expression, (**E**) Transcription factors, (**F**) NFκΒ inhibition (**G**) IFN signaling suppression, and (**H**) Suppression of RIG-I-like sensors. HCA employs a color scheme to represent the extent of gene expression, with gradients ranging from green (suppressed or decreased) to red (increased). More information about the DGEs of these genes can be found in Appendix A.

**Figure 7 pathogens-12-00896-f007:**
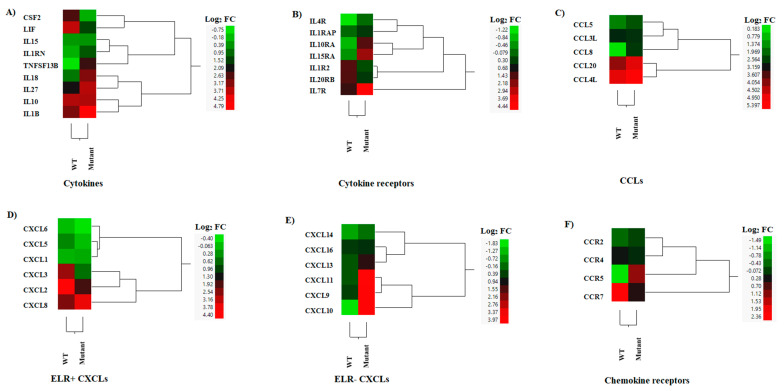
Hierarchical cluster analysis (HCA) showing the differential gene expressions (DGEs) between rNJ0612NME6 (WT) or rNJ0612NME6-M51R (mutant) of genes associated with interleukins, chemokines, and their receptors. These analyses included genes implied with (**A**) Cytokine production, (**B**) Cytokine receptors, (**C**) C-C motif chemokine ligand genes (CCLs), (**D**) Pro-inflammatory cytokines containing a N-terminal glutamate, leucine, and arginine tripeptide motif (ELR+CXCLs), (**E**) ELR-CXCLs, and (**F**) Chemokine receptors. HCA employs a color scheme to represent the extent of gene expression, with gradients ranging from green (suppressed or decreased) to red (increased). More information about the DGEs of these genes can be found in Appendix A.

**Figure 8 pathogens-12-00896-f008:**
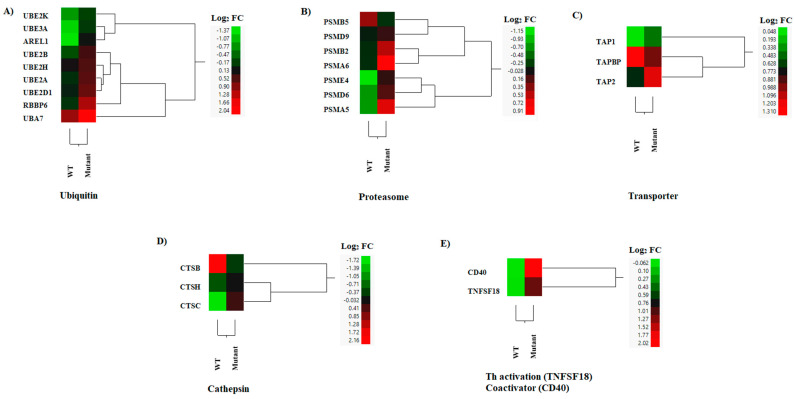
Hierarchical cluster analysis (HCA) showing the differential gene expressions (DGEs) between rNJ0612NME6 (WT) or rNJ0612NME6-M51R (mutant) of genes associated with MHC antigen presentation. Class I antigen processing (**A**–**C**), class II antigen processing (**D**), and antigen presentation (**E**). HCA employs a color scheme to represent the extent of gene expression, with gradients ranging from green (suppressed or decreased) to red (increased). More information about the DGEs of these genes can be found in Appendix A.

**Figure 9 pathogens-12-00896-f009:**
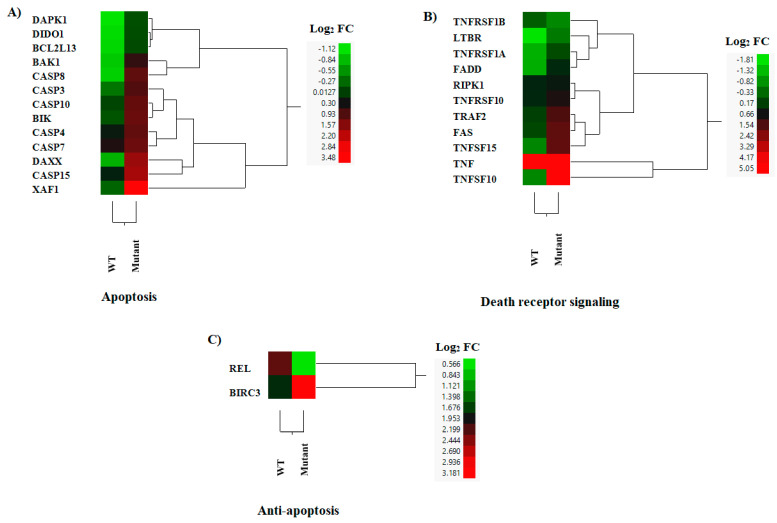
Hierarchical cluster analysis (HCA) showing the differential gene expressions (DGEs) between rNJ0612NME6 (WT) or rNJ0612NME6-M51R (mutant) of genes associated with apoptosis. These analyses included genes associated with; (**A**) Apoptosis, (**B**) Death receptor signaling and (**C**) Anti-apoptosis. HCA employs a color scheme to represent the extent of gene expression, with gradients ranging from green (suppressed or decreased) to red (increased). More information about the DGEs of these genes can be found in Appendix A.

**Table 1 pathogens-12-00896-t001:** The inferred effects of M51R mutation in VSV matrix protein on virus attenuation, vaccine vector, and cancer therapy of VSV based on genes differentially expressed between mutant- and WT-infected cells or genes induced by the mutant infection. Arrows ↑ and ↓ represent increased and decreased expression of specific genes respectively.

Application	Target	Effects Inferred from Differential Gene Expression
Virus attenuation	Non-infected tissues	Inhibiting viral infection: ↑ IFNA, IFNB, IFNL, TNFSF10
		Fever suppression: ↑ IL1RN
		Less severe anorexia: ↓ LIF
	Infected tissues	Less neutrophil infiltration: ↓ ELR+ CXCLs
		Anti-inflammation: ↑ IL1RN, IL10RA
		Killing infected cells: ↑ TNFSF10, TNFSF15, TNFR1A, DR5, FAS, ↓TNFRSF1B
Vaccine vector	Host	Early onset of protection: ↑ IFNA, IFNB, IFNL, TRAIL, TNFSF15
	Antigen presenting cells	Increasing antigen processing: ↑ RBBP6, UBA7, UBE2D1, UBE2H
		Increasing viral peptide MHC-I loading: ↑ TAP1, TAP2
		Enhancing cell-mediated immune response: ↑ IL18
		Enhancing Th1 response: ↑ CD40
		Enhancing Th2 response: ↑ CCL8, TNFSF13B
		Enhancing Th17 and mucosal immune response: ↑ CCL20
		Enhancing Th17, Tfh, Th9 response: ↑ TNFSF18
		Enhancing Tc–APC interactions: ↑ CCLs (4,5)
		Suppressing Treg response: ↑ CD40
		Enhancing Treg response: ↑ IL27
Cancer therapy	Cancer cells	Promoting cancer cell death: ↑ IFNα, IFNβ, IFNλ
		Inducing cancer cell death: ↑ TNF TNFSF10 TNFSF15

## Data Availability

Raw data and the statistical results of the microarray analysis conducted in this study are available at the NCBI Gene Expression Omnibus (GEO) database on the following accession: ID GSE225798.

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
