# Peer review of "Exploring the Molecular Basis of Vesicular Stomatitis Virus Pathogenesis in Swine: Insights from Expression Profiling of Primary Macrophages Infected with M51R Mutant Virus"

_pathogens, 2023, doi:10.3390/pathogens12070896_

Round 1
Reviewer 1 Report
The mutation (M51R) in the matrix protein of VSV, displayed attenuated phenotypes in vivo and in ex-vivo pig macrophages and 14 pigs. To uncover the molecular mechanisms underlying this attenuation, microarray analyses was used to compare gene expression between primary porcine macrophages infected with the M51R virus (rNJ0612NME6-M51R) or its parental virus (rNJ0612NME6). Differentially expressed genes (DEGs) between the two groups were analyzed. The results was some interest to the readers and shed light on the molecular mechanisms underlying VSV infection and attenuate.
The following experiments must be supplemented before this article is accepted.
Some of the DGEs that authors consider them important and meaningful muse be verified by qPCR or/and western blot.
Minor editing of English language required
Author Response
We like to thank the reviewer for the valuable comments to improve the quality of our manuscript. Below you will find our answers to your comments.
Reviewer:
The mutation (M51R) in the matrix protein of VSV, displayed attenuated phenotypes in vivo and in ex-vivo pig macrophages and 14 pigs. To uncover the molecular mechanisms underlying this attenuation, microarray analyses was used to compare gene expression between primary porcine macrophages infected with the M51R virus (rNJ0612NME6-M51R) or its parental virus (rNJ0612NME6). Differentially expressed genes (DEGs) between the two groups were analyzed. The results was some interest to the readers and shed light on the molecular mechanisms underlying VSV infection and attenuate.
The following experiments must be supplemented before this article is accepted.
Some of the DGEs that authors consider them important and meaningful muse be verified by qPCR or/and western blot.
Answer: As requested by the reviewer a validation by qPCR was provided in supplementary file 1.
Reviewer 2 Report
The study by Velazquez-Salinas et al. investigates transcriptional signatures from porcine macrophages infected with the attenuated recombinant VSV strain rNJ0612NME6-M51R, compared to the parental virulent WT strain (rNJ0612NME6). The main objective was to identify immune molecular pathways associated with virulence. Despite in general terms the study is well performed (but with a potential major limitation regarding the number of biological replicates, to be clarified below), for non-specialist audience, in the text it is not clear which is the importance of the work, neither the novelty. In addition, lack of further analyses to validate and explore the hypothesis generated, results in a merely descriptive work.
Major comments:
1. It is not clear which is the novelty of the study. There are several citations (in introduction and discussion) already demonstrating the role of M protein in suppressing IFN-I responses, the main conclusion of the present work. In what sense these results represent a step further in this issue?
2. The abstract is not self-explanatory. They should mention the VSV in the first sentence (for readers not experts on this virus, the name of the strain is not enough). The results from the transcriptomic analyses are not well described. Authors should indicate what pathways/genes are representative of the attenuated versus virulent virus strains. It would also help in the last paragraph of the introduction.
3. An explanation of why the authors decided to analyze samples at 5h post-infection is missing.
4. Can the authors confirm that the experiment has been done with a unique biological replicate (with technical triplicates)? If so, this is a major limitation of the study. Validation of main results as the IFNA ELISA mentioned (but not showed in a figure) in line 274, should be performed in several biological replicates to give consistence to all main conclusions of the study (not only for IFNA).
5. A figure comparing the number of DEGs for each group (virus) would be informative (for instance: number of DEGs, volcano plots and Venn diagrams).
6. Results showed in Tables would be easier to visualize if represented as Heat Maps, as in Figure 4.
Author Response
We like to thank the reviewer for the valuable comments to improve the quality of our manuscript. Below you will find our answers to your comments.
The study by Velazquez-Salinas et al. investigates transcriptional signatures from porcine macrophages infected with the attenuated recombinant VSV strain rNJ0612NME6-M51R, compared to the parental virulent WT strain (rNJ0612NME6). The main objective was to identify immune molecular pathways associated with virulence. Despite in general terms the study is well performed (but with a potential major limitation regarding the number of biological replicates, to be clarified below), for non-specialist audience, in the text it is not clear which is the importance of the work, neither the novelty. In addition, lack of further analyses to validate and explore the hypothesis generated, results in a merely descriptive work.
Major comments:
- It is not clear which is the novelty of the study. There are several citations (in introduction and discussion) already demonstrating the role of M protein in suppressing IFN-I responses, the main conclusion of the present work. In what sense these results represent a step further in this issue?
Answer: We agree with your comment. As expressed in the final sentence of the introduction section the main goal of this study was to identify DEGs and associated biological pathways that may provide insights into the molecular basis mediating the virulence of VSV in pigs. In this sense, we decided to change the title to make it more consistent with the nature of our study “Exploring the molecular basis of Vesicular Stomatitis Virus Pathogenesis in Swine: insights from expression profiling of primary macrophages infected with M51Rmutant virus” Also, we change the legend of figure 1 to make it more explicit. Here we introduce the concept of the use of the M51R mutant as a potential model to get more insights about the molecular basis of the pathogenesis of VSV in pigs. Also, the main conclusion, and the abstract was modified. Furthermore, a statement about the limitations of this study (lack of further analyses to validate and explore the hypothesis generated, results in a merely descriptive work) was included in the discussion section.
- The abstract is not self-explanatory. They should mention the VSV in the first sentence (for readers not experts on this virus, the name of the strain is not enough). The results from the transcriptomic analyses are not well described. Authors should indicate what pathways/genes are representative of the attenuated versus virulent virus strains. It would also help in the last paragraph of the introduction.
Answer: Following the recommendation of the reviewer and considering the limitation in word counting from the Journal we modified the abstract to improve the presentation of our findings.
- An explanation of why the authors decided to analyze samples at 5h post-infection is missing.
Answer: An explanation was added in the methods section. Basically, it was supported by experiments conducted in our previous studies, showing the rapid gene expression in the cells after the infection with a high MOI of VSV. Also, we determined it as an optimal time to avoid the CPE in infected macrophages.
Velazquez-Salinas, L., Naik, S., Pauszek, S.J., Peng, K.W., Russell, S.J., Rodriguez, L.L., 2017. Oncolytic Recombinant Vesicular Stomatitis Virus (VSV) Is Nonpathogenic and Nontransmissible in Pigs, a Natural Host of VSV. Hum Gene Ther Clin Dev 28, 108-115.
Velazquez-Salinas, L., Canter, J.A., Zhu, J.J., Rodriguez, L.L., 2021. Molecular Pathogenesis and Immune Evasion of Vesicular Stomatitis New Jersey Virus Inferred from Genes Expression Changes in Infected Porcine Macrophages. Pathogens 10
Velazquez-Salinas, L., Pauszek, S.J., Holinka, L.G., Gladue, D.P., Rekant, S.I., Bishop, E.A., Stenfeldt, C., Verdugo-Rodriguez, A., Borca, M.V., Arzt, J., Rodriguez, L.L., 2020. A Single Amino Acid Substitution in the Matrix Protein (M51R) of Vesicular Stomatitis New Jersey Virus Impairs Replication in Cultured Porcine Macrophages and Results in Significant Attenuation in Pigs. Front Microbiol 11, 1123.
- Can the authors confirm that the experiment has been done with a unique biological replicate (with technical triplicates)? If so, this is a major limitation of the study. Validation of main results as the IFNA ELISA mentioned (but not showed in a figure) in line 274, should be performed in several biological replicates to give consistence to all main conclusions of the study (not only for IFNA).
Answer: The experiments were conducted using independent biological replicates, using macrophages collected from three independent pigs. This information was clarified in the manuscript. As validation by qPCR was provided in supplementary file 1
- A figure comparing the number of DEGs for each group (virus) would be informative (for instance: number of DEGs, volcano plots and Venn diagrams).
Answer: A comparisons between viruses regarding the number of DEGs associated with the different pathways identified in this study are presented in figure 4. Consistent with it, results from tables were presented in as heat maps.
- Results showed in Tables would be easier to visualize if represented as Heat Maps, as in Figure 4.
Answer: there were presented as heat maps and tables were moved to supplementary material.
Round 2
Reviewer 2 Report
I consider the manuscript can be accepted in the present form